# Three Level Recognition Based on the Average of the Phase Differences in Physical Wireless Parameter Conversion Sensor Networks and Its Effect to Localization with RSSI

**DOI:** 10.3390/s23063308

**Published:** 2023-03-21

**Authors:** Toshi Ito, Masafumi Oda, Osamu Takyu, Mai Ohta, Takeo Fujii, Koichi Adachi

**Affiliations:** 1Department of Electrical & Computer Engineering, Shinshu University, Nagano 380-8553, Japantakyu@shinshu-u.ac.jp (O.T.); 2Department of Electronics Engineering and Computer Science, Fukuoka University, Fukuoka 814-0180, Japan; 3Advanced Wireless & Communication Research Center, University of Electrocommunications, Tokyo 182-8585, Japan

**Keywords:** PhyC-SN, frequency offset, window function, SVM

## Abstract

In recent years, there have been increased demands for aggregating sensor information from several sensors owing to the spread of the Internet of Things (IoT). However, packet communication, which is a conventional multiple-access technology, is hindered by packet collisions owing to simultaneous access by sensors and waiting time to avoid packet collisions; this increases the aggregation time. The physical wireless parameter conversion sensor network (PhyC-SN) method, which transmits sensor information corresponding to the carrier wave frequency, facilitates the bulk collection of sensor information, thereby reducing the communication time and achieving a high aggregation success rate. However, when more than one sensor transmits the same frequency simultaneously, the estimation accuracy of the number of accessed sensors deteriorates significantly because of multipath fading. Thus, this study focuses on the phase fluctuation of the received signal caused by the frequency offset inherent to the sensor terminals. Consequently, a new feature for detecting collisions is proposed, which is a case in which two or more sensors transmit simultaneously. Furthermore, a method to identify the existence of 0, 1, 2, or more sensors is established. In addition, we demonstrate the effectiveness of PhyC-SNs in estimating the location of radio transmission sources by utilizing three patterns of 0, 1, and 2 or more transmitting sensors.

## 1. Introduction

In recent years, wireless sensor networks (WSNs) have become popular because of the development of wireless communication technology. Consequently, the present time is now considered the Internet of Things (IoT) era [1]. In addition, sensor networks have attracted attention from both academia and industry, and the number of deployed sensors has increased exponentially [2,3,4]. WSNs, which collect and utilize diverse sensor information from multiple sensors, have shown tremendous potential for various applications such as security and safety monitoring, disaster management and prevention, industrial automation, traffic management, health management, and smart appliances [5]. In these applications, WSNs must recognize events in a short period of time and take countermeasures during the occurrence of events that cause environmental changes [6,7,8]. Thus, to realize these applications, access control technology that enables WSNs to aggregate information reliably and in a short time is essential to realize these applications.

In packet access, which is an access control technique for WSNs, information is lost because of packet collisions owing to simultaneous access by many sensors [9]. However, carrier-sense multiple access (CSMA), which avoids simultaneous access by detecting multiple potential wireless accesses, has been proposed [10]. Nevertheless, in the case of the widespread deployment of WSNs, packet collisions may still occur because of the hidden terminal problem, wherein the carrier sense cannot detect the other transmitting terminal [11]. In addition, the latency required to avoid packet collisions causes a delay in information aggregation. Meanwhile, non-orthogonal multiple access (NOMA) [12] uses the received signal power difference and interference cancellation. However, stable information aggregation is difficult when the received power fluctuates owing to multipath fading and shadowing.

This study focuses on physical wireless parameter conversion sensor networks (Phyc-SNs) [13]. In this access control technique, a receiver analyzes the frequency spectrum of the received signal by using a fast Fourier transform (FFT). Subsequently, it estimates the transmitted sensor information based on its center frequency. Because the carrier wave has a narrow bandwidth frequency spectrum, the receiver can still detect each carrier wave as a different frequency component if multiple sensors transmit different types of information. This enables simultaneous recognition of sensor information from multiple sensors. However, the receiver cannot identify the source of the information because the carrier wave does not contain an identifier for the sensor. Therefore, the use of a PhyC-SN enables the distribution of sensor information, which is the relative relationship of sensor information notified simultaneously. Moreover, a method has been proposed that uses the distribution of sensor information obtained by a PhyC-SN to position the source of radio emission [14]. Consequently, a PhyC-SN can be used to recognize the distribution of the entire sensor information at low latency, and it can achieve both low-latency sensor information aggregation and a high aggregation success rate.

However, if multiple sensors attempt to notify the same information via a PhyC-SN, each sensor transmits the same carrier wave. In this case, the receiver must detects the number of sensors. The receiver can utilize the received signal energy to estimate the number of simultaneously accessed sensors. A method exists to identify the number of transmission sources based on the evaluation of the amount of energy at each frequency and setting a threshold value according to the number of sensors [15]. However, if the received signal power fluctuates because of multipath fading, the identification accuracy deteriorates significantly. Consequently, a method has been proposed to identify the distribution of sensor information using an on-off decision, which indicates when one or more sensors have been notified, without identifying the number of sensors [14]. However, this method lacks sensor information, which results in a deterioration in the accuracy of sensor information distribution. Therefore, there is a need for a method capable of identifying sensor information, even when multiple sensors report the same sensor information.

In this study, we established a collision detection method for PhyC-SNs that can precisely identify cases where more than one sensor reports the same sensor information. When each sensor is equipped with an inexpensive radio frequency (RF) radio, a frequency difference occurs between the transmitter and the receiver in the local oscillator. This is referred to as the frequency offset [16] and is unique for each sensor. If a single sensor transmits a carrier wave, the phase varies owing to multipath fading. The carrier wave is detected twice consecutively at fixed time intervals to detect the phase difference. Thus, this process is equivalent to a delayed detection. However, if fading does not vary between the two detection intervals, the phase variation due to fading is eliminated, and only that due to the frequency offset can be detected. In general, this process is similar to the method used to estimate the frequency offset using delayed detection [17]. As the estimated frequency offset is independent of fading, it determines the phase variation, which can be estimated from the received signal at each antenna. Moreover, the phase variation remains the same even when multiple receiving antennas are arranged, such that the fading correlation remains uncorrelated. However, when more than one sensor transmits the same carrier wave, one carrier wave is subject to interference from another carrier wave at the receiver. When received simultaneously by multiple receiver antennas, the amplitude and phase variations owing to such interference are different, as the effects of fading are independent. Consequently, regardless of the phase difference estimated by delay detection, each receiving antenna has a different phase difference owing to interference. Although the phase difference in delay detection at each antenna is different when more than one sensor transmits the same carrier wave, the same phase difference can be detected at each receiving antenna in the case of transmission by a single sensor. The variance value, which is the average of the absolute value of the phase change between the receiving antennas, is larger when two or more sensors are transmitted on the same channel than when only one sensor is transmitting. Therefore, the phase-variance value of the received signal between the receiving antennas can be used as an identifier to detect collisions. Thus, this study establishes a method to identify whether one, two, or more sensors access the same channel based on the phase dispersion values of multiple antennas coupled with the amount of received power. We evaluate the effectiveness of the proposed method and show that it can achieve a collision detection accuracy that is higher than that of the conventional energy detection method [15]. Furthermore, the method was applied to the PhyC-SN radio wave monitoring system and achieved better detection accuracy than the conventional method [14] in terms of the location estimation of the radio wave source.

The contributions of this study are as follows: 1. A scheme based on the average phase of differences for distinguishing between two or more accessing sensors, which is equivalent to the collision detection of simultaneous access from two or more sensors, is proposed. 2. To improve the detection accuracy, we propose antenna selection, long-term window function, and two-dimensional detection based on a support vector machine. 3. When we used a PhyC-SN with the proposed three-level detection for gathering the RSSI measured by the radio sensor, we evaluated the accuracy of the localization. We clarified the effect of the proposed scheme in terms of the localization accuracy.

The basic principle of the proposed method was presented at an international conference [18], wherein the improvement in aggregation efficiency through the suppression of the effect of intercarrier interference using long-interval FFT was clarified. However, the identification accuracy of the collision detection identifier using the phase difference fluctuations due to noise was not discussed. Thus, to clarify the issue of noise robustness, we enhanced the detection accuracy by improving antenna selection. The application of information aggregation in PhyC-SNs has not yet been discussed. In this study, we demonstrated that when using PhyC-SNs to estimate the location of a radio transmission source [14], the proposed method improves information aggregation and estimation accuracy.

## 2. Related Studies

Theoretical analysis using the statistical trends of signals has been presented as a signal detection method [15]. In [15], a multistate detection method was proposed; however, the identification accuracy deteriorated with an increase in the number of detected states. In CSMA, an access method using collision detection was considered [19]. For collision detection, signals from other sensors were detected using the access suspension period. In addition, assuming frequency sharing, a method using the guard time of time-division duplexing (TDD) has been proposed to detect other signals during access by the system [20]. However, both methods require a certain stop time, which increases the detection time. Thus, a method for detecting the stop time is necessary.

Energy [21], matched filters [22], and periodic stationarity detection [23] have been proposed as methods for capturing signal features. However, the accuracy of energy detection deteriorates with the occurrence of power fluctuations, owing to multipath fading. In particular, when multiple signals are detected, the power difference according to the number of sensors decreases owing to fading, which results in a significant degradation in the detection accuracy. Matched filter detection does not improve the detection sensitivity because of poor matching accuracy in environments where multiple signals are mixed. Periodic stationarity detection exhibits good performance when multiple signals are mixed, and there exists an inherent periodicity; however, the generation of signals with different periodicities depending on the number of signals has not been considered.

In a nonorthogonal multiplexing scheme [12], where individual signals are detected via interference cancellers and maximum likelihood detection methods in an environment accessed by many terminals, several methods for detecting the presence of signals have been studied. For signal detection before canceller operation, a signal detection method using a support vector machine (SVM) [24] and an active user detection method using the alternating direction method of multipliers (ALM) [25] have been proposed. These methods are based on the premise that the modulation of the transmitted signal does not affect signal detection. Because a PhyC-SN is an unmodulated signal, its features are extremely limited, and the derivation of features for using these methods remains challenging.

In addition, the RF fingerprint, wherein the unique signal fluctuation caused by RF individual differences is identified, has been considered [26]. To realize the RF fingerprint, an RF signal fluctuation unique to each sensor is required; however, the detection of stable signals is challenging owing to the uncertainty of the RF signal. The arrival wave estimation method based on null beam formation in an adaptive array antenna enables the individual separation of simultaneously accessed signals [27]. However, the number of antennas required increases in the case of many signals, resulting in a considerable increase in the complexity of the receiver.

In signal detection in a PhyC-SN, which is the focus of this study, the transmitted signal is unmodulated; thus, the signal possesses few features. Therefore, to improve detection accuracy, identifiers that vary with the number of sensors must be established. In situations where unmodulated signals are mixed, this method detects signal mixing. However, to the best of our knowledge, there are no methods for detecting collisions; thus, a method using phase differences established in this study is proposed as a new method.

We consider the specification of the information source based on a particular frequency offset in a PhyC-SN [28]. However, the proposed scheme requires time continuity of the sensing information; thus, it is not available in the initial data gathering or non-time continuous sensing data. We also considered the application of PhyC-SNs to localization [14]. To counter the impact of multipath fading, on-off detection is proposed. When two or more sensors send the same sensing data to the receiver via the PhyC-SN, the sensing data are deleted. This resulted in localization errors [14]. In this paper, to avoid the deletion of sensing data, we propose a three-level detection, which is none, informed by a sensor, and informed by two or more sensors. In addition, to improve the detection accuracy, antenna selection, long-term window, and two-dimensional detection based on a support vector machine are considered.

## 3. Physical Wireless Parameter Conversion Sensor Networks(PhyC-SNs)

In this study, we assume a radio sensor that observes the received signal strength indicator (RSSI) of radio waves [8]. Each sensor notifies the aggregation station of the RSSI observed as sensing data. Figure 1 presents an overview of the information notified by a PhyC-SN. The RSSI is quantified at regular intervals. The *m*th quantized RSSI value is $p_{m}$. Let $m=1,2,\ldots ,N_{r}$, where $N_{r}$ denotes the total number of quantized RSSI values. In this study, the available bandwidth was divided, and multiple channels were set up. The total number of channels was equal to the total number of quantized RSSI values $N_{r}$, where the PhyC-SN correspondence table assigns the *m*th quantization number to the *m*th channel. A correspondence table was created relating the channel number and the quantized sensing data and was shared between the transmitter and receiver before communication began.

Each sensor transmits a carrier wave with the frequency of the channel number corresponding to the sensing data. An inverse fast Fourier transform (IFFT) was used to generate the carrier wave, which was equivalent to an OFDM subcarrier wave and was unmodulated.

At the receiver, the carrier waves transmitted by all the sensors were detected over a fixed time length. The frequency spectra of all the channels were detected simultaneously using a fast Fourier transform (FFT) of the detected signals. A threshold value was set for the power dimension of the frequency spectrum for each channel. When the power of the frequency spectrum exceeded the threshold value, the sensor was identified as transmitting a carrier wave through that channel. The RSSI reported by the sensor can be recognized from the correspondence table between the channel number and sensing data.

However, if multiple sensors detect the same RSSI quantization number and transmit the same channel carrier, multiple power thresholds are established to identify the number of sensors. This was realized by exploiting the tendency of the average power to increase in proportion to the number of carrier waves. However, because the ID that identifies the sensor is not assigned to the notified RSSI, identifying the sensor that notifies the RSSI is impossible. Therefore, when a PhyC-SN is used, the receiver can identify the frequency distribution of the RSSI for all sensors.

Figure 2 shows a case of radio signal notification by a PhyC-SN in an actual wireless transmission channel. The received power fluctuated owing to multipath fading. In addition, because the carrier waves of multiple sensors were accessed asynchronously, carrier waves of the same frequency were asynchronously synthesized at the receiver. Consequently, power fluctuations occurred in the composite signal. These power fluctuations caused sensor identification errors in the threshold judgment of the receiver, which misidentifies the number of sensors from the actual number of sensors.

Another cause of misidentification of the number of sensors is frequency offset. The frequency offset is the difference in the center frequency of the transmitter between the transmitter and receiver [16]. They are particularly likely to occur in the inexpensive radios used for sensors. When a frequency offset occurs, the power of the carrier wave is reduced, and there is a leakage of power to adjacent channels [28]. In the case of leakage in a channel with no station-emitted carrier waves, false detection occurs, where the carrier wave is recognized as having been transmitted from a channel that has not actually emitted any waves. However, the occurrence of leakage into a channel where a carrier wave has been transmitted results in power fluctuations in the carrier wave, leading to the misrecognition that a carrier wave does not exist.

In a PhyC-SN, multipath fading, asynchronous compositing between carriers, and frequency offsets result in identification errors in the number of carriers (which is equivalent to the number of carriers transmitted by the sensor).

## 4. Proposed Sensor Number Identification Method Using PhyC-SNs

In this study, we established a method for identifying three states in each channel. The first was when no sensor transmitted a carrier wave, that is, the number of sensors was 0. The second was when one sensor announces a carrier wave, and the number of sensors was one. The third was when the number of sensors was two or more, with two or more sensors notifying the carrier wave. We defined a collision as a situation where the number of sensors was two or more because this is a state in which multiple sensors access the same channel simultaneously. Therefore, our proposed method was a collision-detection method that identified cases when the number of sensors was two or more.

### 4.1. Collision Detection Method Using the Average of Phase Differences

The collision-detection method proposes a new identifier that exploits the frequency offsets and uses both the amount of energy and the proposed identifier to detect collisions.

Figure 3 shows the frequency offset estimation process for a single channel. This process is similar to delay detection, wherein a complex conjugate is applied to the received signal after one symbol interval, and is multiplied by the received signal. In a PhyC-SN, an unmodulated carrier wave is transmitted. If a frequency offset, which is the frequency difference between the receiver and transmitter, occurs at each sensor and receiver, the phase of the received carrier wave is shifted. Therefore, the frequency offset can be estimated equivalently by estimating the phase transition in one symbol interval using delay detection [16].

Figure 4 shows the transmitter processing using the proposed method. On the transmitter side, the received sensing data are mapped to the corresponding channel, and IFFT is performed. The signal in one symbol section obtained by IFFT was duplicated, and multiple symbols were transmitted by frequency upconversion.

Next, the characteristics of the amount of phase transition corresponding to the number of sensors that select the same channel are explained.

Consider the case where the *i*th sensor transmits a carrier wave on a certain channel and detects the signal at the *k*th receiving antenna from time *t*.

The received signal $A_{k}$ and the received signal $B_{k}$ delayed by one symbol time are described as follows:(1)$$\begin{array}{c} A_{k}=h_{i,k}e^{-j\omega_{i}t} \end{array}$$(2)$$\begin{array}{c} B_{k}=h_{i,k}e^{-j\omega_{i}\left(t+T\right)}, \end{array}$$
where *T* is one symbol length, $\omega_{i}$ is the frequency offset at the *i*th sensor, and $h_{i},k$ is the fading coefficient of the *k*th receiving antenna at the *i*th sensor. The station-generating transmitters of each receiving antenna were synchronized using wired or other techniques. Therefore, the received signals of each receiving antenna were assumed to have the same frequency offset magnitude. The transmitted signal did not influence this process and was omitted for simplicity. In addition, noise is omitted to simplify the equation.

The following equations were obtained by multiplying the received signals $A_{k}$ and $B_{k}$ by the two variables to which the complex conjugation was applied.
(3)$${\hat{\theta }}_{k}=\arg\left(A\times B^{*}\right)=\arg(|h_{i,k}{|}^{2}e^{j\omega_{i}T}),$$
where ${\hat{\theta }}_{k}$ is the amount of phase transition estimated by delayed detection at the *k*th receiving antenna.

In the case of a single sensor, the phase components of ${\hat{\theta }}_{k}$ and $h_{i,k}$ are canceled by the delay detection. Therefore, the phase components are estimated to be the same for each receiving antenna.

Next, the case in which the two sensors transmit carrier waves in the same channel is described.

The received signal $A_{k}$ and the received signal $B_{k}$ delayed by one symbol are described as follows:(4)$$A_{k}=h_{i,k}e^{-j\omega_{i}t}+h_{l,k}e^{-j\omega_{l}t},$$
(5)$$B_{k}=h_{i,k}e^{-j\omega_{i}\left(t+T\right)}+h_{l,k}e^{-j\omega_{l}\left(t+T\right)},$$
where is the amount of frequency offset $\omega_{l}$ at the *l*th sensor, and is the fading coefficient $h_{l,k}$ for the *k*th receiving antenna at the *l*th sensor. The carrier waves of the *i*th and *l*th sensors are synthesized at the receiver.

The received signals $A_{k}$ and $B_{k}$ were multiplied by two variables coupled with complex conjugation. The result of the delay-detection process is
(6)$$\begin{array}{c} {\hat{\theta }}_{k}=\arg\left(A\times B^{*}\right)=\arg(|h_{i,k}{|}^{2}e^{j\omega_{i}T}+h_{i,k}h_{l,k}e^{j\omega_{i}\left(t+T\right)}e^{-j\omega_{l}t}\;\;\;\;\;\;\;\;\;\\ +h_{i,k}h_{l,k}e^{-j\omega_{i}t}e^{j\omega_{l}\left(t+T\right)}+|h_{l,k}{|}^{2}e^{j\omega_{l}T}). \end{array}$$

When two sensors transmit carrier waves on the same channel, phase fluctuations due to fading appear in the second and third terms of the above equation. If the fading coefficients of each receiving antenna are statistically independent, then the phases of the phase transients obtained at each antenna are different. Thus, the phase of the phase transition of the composite signal varies from antenna to antenna owing to mutual interference between carrier waves. Therefore, if the phase transition obtained at each receiving antenna exhibited the same phase, it could be assumed that only one sensor transmitted a carrier wave on that channel. However, if the phase of the phase transition was different at each receiving antenna, then it could be assumed that two or more sensors transmitted a carrier wave on that channel, and we could assume that a collision had occurred.

In this study, to detect the presence of phase differences between antennas, the average of the following phase differences was derived as an identifier for collision detection.

Figure 5 shows the phase difference between the points estimated by the two antennas using the delayed detection. The blue points in Figure 5 are the estimated points, and the phase difference between the received points is indicated in red. The phase difference is always calculated at an inferior angle and does not exceed 180 degrees.

The phase difference is expressed as follows:(7)$$\omega =\frac{1}{2}\sum_{i=1}^{n}\sum_{j=1}^{n}{\left({\hat{\theta }}_{i}-{\hat{\theta }}_{j}\right)}^{2}$$
$${\hat{\theta }}_{a}-{\hat{\theta }}_{b}=\left\{\begin{array}{c} {\hat{\theta }}_{a}-{\hat{\theta }}_{b}\;\;\;\left({\hat{\theta }}_{a}-{\hat{\theta }}_{b}<180\right)\\ 360-{\hat{\theta }}_{a}-{\hat{\theta }}_{b}\left({\hat{\theta }}_{a}-{\hat{\theta }}_{b}\geq 180\right) \end{array}\right.$$
(8)$$\phi =\frac{\omega }{K(K-1)},$$
where *K* is the number of receiving antennas and ω is the squared value of the phase difference estimated at each receiving antenna and averaged per antenna. The average value of the phase difference is ϕ averaged from ω, and is used as an identifier for collision detection.

If two or more sensors transmitted a carrier wave on the same channel, the phase of the received signal of each antenna fluctuated because of the mutual interference of the carrier waves; thus, the identifier ϕ for collision detection was larger than the case when a single sensor transmitted a carrier wave. Hence, the average value of the phase difference ϕ was the normative value for the collision detection.

### 4.2. Application of a Long-Section Window Function

The frequency offset causes the subcarrier orthogonality to break down, resulting in spectral leakage to the other channels [29]. This spectral leakage causes significant degradation of the detection accuracy [28]. This study considered the application of a long-interval window function as a countermeasure to spectral leakage.

The transmitter transmits a carrier wave over a time interval of multiple symbols. The receiver continuously detects the time intervals of multiple symbols. In an actual wireless transmission channel, the arrival timing of signals differs because of the signal delay spread caused by multipath fading and access timing errors of each sensor. Various schemes of timing synchronization for sensor networks have been considered thus far [30] and these are not our research targets. In this case, by transmitting multiple symbols, the leading and trailing one-symbol intervals perform the same function as the guard interval in OFDM, thereby avoiding the loss of frequency orthogonality owing to symbol-timing errors [31]. For simplicity, we assumed that all signals arrived simultaneously and that there was no delay spread.

At the receiver, the weights of the window function were multiplied by the time widths of the multiple symbols.

Figure 6 shows the difference between short and long intervals. In the long section, an FFT was applied with the number of samples matching the multisymbol time-length.

In the long-interval FFT, there are multiple symbols in the detection signal. Consequently, the periodicity of the carrier wave increased, and the frequency spectrum was concentrated at the carrier frequency. Simultaneously, sidelobes, which spread to other frequencies, were suppressed, thus reducing the spectral leakage owing to frequency offsets. As shown in Figure 6, the application of a window function to each FFT in the short interval causes distortions in the forward and backward directions of each symbol. However, in the case of the long-interval FFT, the signal distortion caused by the window function is limited to the first and last symbols among multiple symbols. Thus, the long-interval FFT reduced the effect of distortion owing to the window function and suppressed the spectral leakage.

### 4.3. Antenna Selection Methods for Improved Detection Sensitivity

Figure 7 shows the received signal points of each antenna in the I-Q plane when one sensor transmits a carrier wave. The magnitude was normalized to one symbol and the average power was set to one. Furthermore, there were no collisions, and the received signal points of all antennas exhibited the same amount of phase transition. However, received signals with low power were susceptible to noise. Therefore, the received signal point with a high power that was tolerant to noise was selected, and the identifier ϕ for collision detection was calculated. The receiver was equipped with multiple received antennas, and thus, multiple received signal points are detected. The sensing data can be detected at any received signal point, and thus the receiver can select the signal point with a large signal power to improve the accuracy of detecting the sensing data. Because each signal point is derived from each receiving antenna, the selection of the signal point is equal to the selection of the antenna. For antenna selection, we used a certain threshold. If the power of the received signal point derived from the receive antenna is greater than the threshold, it is used for detecting the sensing data; otherwise, it is deleted. The larger the selected threshold value, the fewer the number of received signal points exceeding the threshold value. Consequently, the number of signal points for calculating the average phase difference and the noise reduction effect by averaging were reduced. However, for a small threshold, the number of received signal points that exceeded the threshold was reduced. Furthermore, when the threshold was reduced, the phase components were calculated even for received signals with low received power; thus, they were strongly affected by the noise for the collision detection identifier ϕ. Therefore, an appropriate threshold must be set to enhance the effect of noise reduction.

The threshold value of *t* was calculated as follows:

In the case of rectangular windows
(9)t=EW.

For the case of BHW
(10)t=EW/4.

Let *E* be the average power of a received signal. The threshold ratio *W* was determined based on the average power. The application of the window function attenuates the amplitude of the received signal. In this study, the threshold applied to the BHW(Blackman-Harris window) was one-fourth that applied to the rectangular window.

The red circles in Figure 7 indicate the power levels at W=0.5, which were normalized to the average power.

The receiver process, which involves a combination of antenna selection, window function, and phase difference averaging, is shown in Figure 8. The receiver calculated two features for each channel: the energy value and average of the phase difference. Thereafter, these two features were used to identify the number of sensors.

### 4.4. Collision Detection Method

Figure 9 and Figure 10 show the scatter plots of the average values of the energy and phase difference when there are one and two transmitting sensors. Red and blue indicate the detection values when the number of transmitting sensors was one and two, respectively. The average values of the energy and phase differences tended to differ depending on the number of sensors. Therefore, a machine-learning support vector machine (SVM) was used to identify the number of sensors [32]. The various parameters of the SVM were determined through the automatic optimization of hyperparameters in MATLAB provided by MathWorks [33]. Figure 9 and Figure 10 show a scatterplot of the energy values and the average of the phase differences. Here, SVM determines the threshold that distinguishes between one or two sensors.

Figure 9 and Figure 10 show the scatter plots with and without antenna selection, respectively. Here, W=0.5 was applied as the threshold ratio for antenna selection. Antenna selection was considered to improve the identification accuracy because the scatter plots of sensors 1 and 2 were concentrated, and the distance between them increased.

## 5. PhyC-SN Use Case for Estimating the Number of Sensors

Herein, we applied this method to positional fingerprinting, which is an application of PhyC-SNs for estimating the number of sensors. In the position fingerprinting method, sensors that can observe RSSI values are uniformly placed. The radio source was the positioning target. Each sensor measured the RSSI while radio waves were emitted, and the measured RSSI was reported to a central station. The RSSI was then reported to a single aggregation station that used a PhyC-SN as the method for reporting the RSSI from each sensor [14].

### 5.1. Aggregation Method

Figure 11 shows a diagram of the aggregation of sensor information by a PhyC-SN. As in [14], a PhyC-SN defines areas within a certain range of sensor locations, and the sensors in each area are simultaneously notified by the PhyC-SN. However, slots are defined as time divisions, and each area is notified at a different time slot. Let *b* be the number of areas (b∈1,2,…,B), where *B* is the total number of areas.

### 5.2. Calculating the Similarity of Estimation Criteria

The position fingerprinting method has two phases: pretraining and positioning. In the pre-training phase, the locations of the radio transmission sources are known, and the radio source sensors are placed uniformly within the deployed area. Let *l* be the location number of the source in advance ($l\in 1,2,\ldots ,N_{l}$), where $N_{l}$ is the total number of radio transmission sources deployed in the pretraining phase.

The RSSI was quantized at regular intervals, and the quantization number was transmitted corresponding to the subcarrier number [14]. Furthermore, an on-off decision where the receiver compared the power of each subcarrier with a threshold value and determined that the sensor had been notified when the power was above the threshold value was proposed [14]. Although the threshold decision eliminates power fluctuations during propagation, it identifies a single sensor as transmitting a subcarrier wave, even when multiple sensors transmit the same subcarrier wave. This results in the loss of information regarding the number of sensors that have been notified.

Next, the RSSI was obtained from the frequency components of the channels determined to be above a certain threshold value, based on a conversion table. Consequently, the RSSIs detected from the channels above the threshold were sorted in descending order to obtain a feature vector that indicates the characteristics of the radio source. Thus, to compensate for the fact that the number of RSSIs reported by the sensors was reduced by the reception process [14], a feature vector with a size equal to the total number of sensors in the area was generated by adding noise-level RSSIs to the aggregate result. Thus, when the *b*th area was identified using the PhyC-SN, the following RSSI aggregation vector was obtained:(11)$$\begin{array}{c} {\tilde{p}}^{l,b}=\left[{\tilde{p}}_{1}^{l},{\tilde{p}}_{2}^{l},\ldots ,{\tilde{p}}_{N_{b}}^{l}\right]. \end{array}$$

Let ${\tilde{p}}_{b}^{l,b}$ be the RSSI aggregate vector formed when a sensor in the *b*th area notifies the location *l* of the radio source. Furthermore, let $N_{b}$ be the total number of radio sensors in the *b*-th area, and let ${\tilde{p}}_{n}^{l}$ be the *n*th largest RSSI value in the aggregate vector.

The aggregate vector is formed when a sensor in the *b*th area reports an RSSI during the positioning phase.
(12)$$\begin{array}{c} {\tilde{p}}^{*,b}=\left[{\tilde{p}}_{1}^{*},{\tilde{p}}_{2}^{*},\ldots ,{\tilde{p}}_{N_{b}}^{*}\right]. \end{array}$$

In the literature, [14] calculated the squared Euclidean distance between the aggregate vector obtained in the positioning phase and that obtained in the pre-training phase for the location of the radio source. Furthermore, the position of the pre-training point with the shortest distance, $l^{☆}$, was used as the location estimation point. $l^{☆}$ is expressed as follows:(13)$$\begin{array}{c} l^{☆}=\underset{\forall l}{\arg\;\min}\sum_{b=1}^{B}\left({\tilde{p}}^{*,b}-{\tilde{p}}^{l,b}\right){\left({\tilde{p}}^{*,b}-{\tilde{p}}^{l,b}\right)}^{T} \end{array}$$

### 5.3. Proposed Method: Environment Recognition Using 3-Level Detection

In the proposed method, the PhyC-SN can recognize the number of sensors at three levels: 0, 1, and 2 or more. When applying the proposed method to positional fingerprinting, the number of sensors was judged to be two if two or more sensors were recognized. This is because the possibility that there were two sensors was higher than those of other numbers of sensors. In [14], more information on the number of sensors was obtained than in the case of detection with a single-threshold decision.

Figure 12 shows the difference in the aggregated RSSI results between the three-level identification of the proposed method and the conventional on-off identification when the RSSI was notified by the PhyC-SN. The RSSI values received by each sensor are color coded. The histogram shows the RSSI value when it was aggregated by the PhyC-SN from a sensor and the number of sensors that received it. In the conventional method, which is an on-off decision, even if two or more sensors inform the receiver of the same RSSI, the receiver recognizes that one sensor informs the RSSI. Therefore, some RSSI data are lost. In the proposed method, if two sensors provide the same RSSI, the receiver recognizes that the number of sensors forming the RSSI is two. If three or more sensors indicate the same RSSI, it recognizes that the number of sensors forming the RSSI is two. Here, the vector size was maintained even when sensor information was lost via the addition of noise-level RSSI, as in the literature [14]. The proposed method is expected to improve the accuracy of position estimation because less information is lost from the sensors.

## 6. Simulation Evaluation

### 6.1. Numerical Results Part 1 (Aggregate Accuracy of a PhyC-SN)

As in [14], we used data from outdoor sensors and radio-wave sources. The thirty-five radio sensors measuring the RSSI values were placed in an area of approximately 300 m × 300 m. The number of position patterns in the radio transmission source was assumed to be 124 and arranged in a mesh-like manner. We used LoRa aggregation stations in the 920 MHz band as the radio source. However, each sensor was assumed to be a LoRa sensor terminal, and the LoRa terminal periodically transmitted signals including sensor information. The sensor was an LHT65 provided by DRAGINO. The aggregation station was LPS8 provided by DRAGINO. The aggregation station calculates the RSSI from the signals sent by the sensor, which is equivalent to the RSSI of the sensor in the signal sent by the aggregation station owing to the radio wave coverage [34]. From this objectivity, the RSSI observed from the signal transmitted by each sensor was used as the RSSI observed by each sensor, and this RSSI was sent to the aggregation station using a PhyC-SN.

The RSSI was quantized in 6 dB intervals in the range −136 dBm∼−40 dBm, each corresponding to each subcarrier. In this case, an RSSI below −136 dBm was assumed to be below the noise, and sensors below the noise were not transmitted to the aggregation station. There were 16 subcarriers.

### 6.2. Simulation Parameters for Evaluation of the Aggregation Accuracy

We constructed a baseband simulation of the wireless communication using MATLAB. The simulation parameters are listed in Table 1. In a Rayleigh fading environment, the number of receiving antennas was four. The number of subcarriers was set to 16, and the maximum number of sensors to be aggregated was 35. Thus, if all the sensors notified the PhyC-SN, a collision would occur in one of the channels. The frequency offset was modeled as a uniform random number in the range [−0.4 0.4], and the normalized frequency offset was normalized by the channel bandwidth.

Two types of window functions were used: rectangular and BHWs functions. To evaluate this experiment, a method using energy detection was considered the conventional method. Conversely, the proposed method uses energy detection and the average phase difference by the frequency offset.

### 6.3. Aggregate Accuracy Results

Figure 13 shows the threshold ratio *W* and the error rate due to antenna selection. The error rate was defined as the number of errors divided by the number of estimation attempts, where the error was defined as the number of sensors identified as different from the number of sensors that were actually transmitted.

In this figure, we can see that the proposed scheme achieves a smaller error rate than the conventional scheme of energy detection. This is because energy detection suffers from fluctuations in the received signal power caused by multipath fading. In the proposed scheme, the average phase of difference can mitigate the phase distortion caused by multipath fading, and it is effective in distinguishing the collision of accessing multiple sensors.

The figure shows that the error rate tends to be convex downward with a value of *W*. When calculating the average phase difference, a small threshold value selected the antennas receiving signal power with low noise tolerance, thereby degrading the identification accuracy. However, a high threshold value reduces the number of selected antennas and the noise reduction effect of the averaging process. Therefore, there exists an optimal *W* in the window function. In the following evaluation, W=0.5 is used.

In addition, BHW achieved a smaller error rate than the rectangular windows. BHW can suppress the spectrum leak to the other subcarriers, and thus, the error decision caused by the spectrum leak is avoidable.

The error rate for different numbers of transmitted symbols is shown in Figure 14 where a 20 dB SNR was assumed. In this figure, “Energy & phase dispersion detection” uses a short FFT whose size is one symbol. When the number of symbols increases, short FFTs are iteratively performed, and then the detected signal components are combined. In the “Proposal method” and “Proposal method (BHW),” the size of the FFT is equal to the number of symbols, and it is long. In the case of identification with the addition of the mean value of the phase-difference feature, the error rate did not change with a change in the number of transmitted symbols when the FFT size was one symbol. In contrast, in the case of FFT over a long section, the identification accuracy was improved by increasing the number of transmitted symbols, thereby increasing the FFT size, concentrating the frequency spectrum at the center frequency of the carrier wave, and suppressing its spread to other frequencies. When the number of symbols was eight, the rectangular window achieved the lowest error rate, whereas for more than 10 symbols, the BHW achieved the lowest error rate. When the number of transmitted symbols was small, a rectangular window with low noise immunity and a sharp main lobe can increase the detection power and improve the error rate. However, the detection power of the main lobe could be improved by increasing the number of transmitted symbols. Furthermore, the BHW suppressed the sidelobes, which suppressed other frequency spectral leakages, thus resulting in improved identification accuracy.

Figure 15 shows the results of the relationship between the SNR and error rate for different window functions. The number of symbols is set to 20. Regardless of antenna selection, for SNRs lower than 8 dB, the rectangular window was superior because of its weak noise immunity and small power leakage owing to the frequency offset. Conversely, in the case of SNR higher than 8 dB, the BHW was superior because it was more noise tolerant and the power leakage owing to the frequency offset was larger.

Figure 16 presents a comparison of the relationship between the SNR and error rate for conventional energy detection and the proposed method. Although the error rate was lower than that of the conventional method owing to the addition of the average phase difference, the identification accuracy was saturated because the FFT size was one symbol. Therefore, in addition to antenna selection, an FFT was performed over a long interval, which increased the FFT size, and the identification accuracy was improved.

The error rate of the conventional energy detection method saturated at approximately $10^{-1}$ did not improve with SNR expansion. In contrast, the proposed method with BHW achieved an error rate of less than $10^{-2}$ and superior identification accuracy. Furthermore, it suppressed the spectrum leakage through the application of the long interval FFT and BHW, and achieved excellent discrimination performance using a combination of the phase variance and energy amount.

## 7. Numerical Results Part 2 (Positioning Accuracy)

Location estimation using a PhyC-SN used data generated by the Wireless InSite radio propagation analysis tool. The environment for generating data is presented in Table 2. A total of 136 sensors were placed in an urban space measuring 800 m × 800 m. There were eight divisions for the aggregation. To identify the source of the radio waves, 37 points were placed at the source of the radio waves as preliminary training, with the source of the radio waves known. Subsequently, 25 radio transmission sources were placed as location estimation points, and their locations were unknown. The location estimation error is the error between the estimated location points and coordinates derived from the location estimation. The sources used LoRa, which is an LPWAN standard, with a center frequency, bandwidth, and transmission power of 920.6 MHz, 0.125, and 10 dBm, respectively. In addition, they emitted radio waves uniformly in all the directions. This study evaluated the positioning error of the position estimation when the number of channels was aggregated to 16 and 64.

### Position Estimation Error

Table 3 presents the identification errors for the number of sensors when a PhyC-SN aggregates the number of sensors to three levels:0, 1, and 2 or more. A low error rate was achieved by increasing the number of channels. This can be attributed to the increase in the number of channels, which reduces the probability of collisions via the distribution of channels over which the sensors transmit their carrier waves. Regardless of the number of channels, the proposed method achieved identification errors approximately one order of magnitude lower than those of conventional energy detection. Figure 17 shows the cumulative distribution function (CDF) of the positioning error when the number of channels was 16. “Ideal” is the error when the nearest radio source was selected from the radio transmission source used in the pre-training phase of the location fingerprinting method, and represents the upper boundary of the estimation error when using the location fingerprinting method [14]. Furthermore, “Packet Communication” is the case in which information is aggregated using conventional packet access. However, it was assumed that no identification errors occurred during packet access. In “PhyC-SN,” we provide the results of three-level identification using only energy detection and three-level identification using the phase variance of the proposed method. For comparison, we provided positioning results using the on-off detection method [14], which can identify the presence of a sensor.

The figure shows that the on-off detection method yielded a positioning error greater than 100 m in approximately 16% of the cases, whereas the 3-level detection with the PhyC-SN’s energy identification method yielded a positioning error of approximately 8%. Furthermore, three-level detection using the proposed method reduced the error by approximately 1%. Therefore, a high positioning accuracy was achieved using the proposed method. This is because the proposed method can discriminate three-level detections with high accuracy. Moreover, owing to the expansion of the number of discrimination levels, the amount of lost sensor information is reduced, which improves the positioning accuracy. However, the proposed three-level detection method achieved positioning accuracy close to that of “Packet Communication,” although it exhibited a degradation of approximately 0.05 in the CDF. The number of times required for sensor information aggregation in “Packet Communication” and “PhyC-SN” was 136 slots for both, with the access time of one sensor defined as one slot. The number of times required to aggregate sensor information was 136 slots for the former and eight slots for the latter, when the access time of one sensor was defined as one slot [14]. As “PhyC-SN” has an area division of 8, we assumed that one slot was allocated to each area to aggregate sensor information. Consequently, “PhyC-SN” required approximately 1/17 of the time required for aggregation compared to “Packet Communication” thus significantly reducing the aggregation time. Therefore, “PhyC-SN” and the proposed method achieved positioning accuracy approaching that of “Packet Communication” while achieving fast positioning. Figure 18 shows the CDF of the positioning error when the number of channels was 64.

The figure shows that “Packet Communication” and “PhyC-SN” with the proposed method achieved approximately the same positioning accuracy when CDF = 0.9 or higher. This can be attributed to the improvement in identification accuracy due to increasing the number of channels coupled with the reduction in the probability of three or more sensors selecting the same channel simultaneously owing to the increase in the number of channel selections by the sensors. This reduces the possibility of loss of sensor information. Therefore, the proposed method can achieve a positioning accuracy equivalent to that of "Packet Communication" and fast positioning by allowing the required frequency bandwidth to increase owing to the increased number of channels.

## 8. Conclusions

In this study, we propose a new identifier, referred to as the phase variance value, focusing on the sensor-specific frequency offset, to identify the number of sensors that select the same channel in the PhyC-SN method. In addition, to improve the accuracy of identification, we established a long-interval fast Fourier transform and a window function for signal detection. Consequently, an antenna selection method based on a detection-power criterion was proposed. The evaluations performed through computer simulations showed that the proposed method could discriminate the number of sensors with higher accuracy than the conventional energy detection method, which discriminated the number of sensors as 0, 1, and 2 or more. As an application of PhyC-SNs, we applied the proposed method to the position fingerprinting method, which measures the position of a radio source using the observation results of many sensors. A PhyC-SN coupled with the proposed method achieved better positioning accuracy than the conventional energy detection method owing to the improved accuracy in identifying the number of sensors. Moreover, the proposed PhyC-SN achieved a positioning accuracy close to that of the packet-based wireless access method and significantly reduced the time required for complete positioning. Therefore, the proposed method achieves fast positioning.

As the number of sensors and subcarriers in the proposed method increases, the computational process of pre-training becomes more time consuming. Therefore, it is necessary to reserve the time for pretraining when using the proposed method on an actual embedded platform. Because the propagation state changes over time, the number and timing of the pre-training should be investigated. These are important topics for future research.

Currently, identification is possible when there is no access, and the number of sensors is one, two, or more. Consequently, the improvement of the identification method for a larger number of sensors is also an important issue for future research.

## Figures and Tables

**Figure 1 sensors-23-03308-f001:**
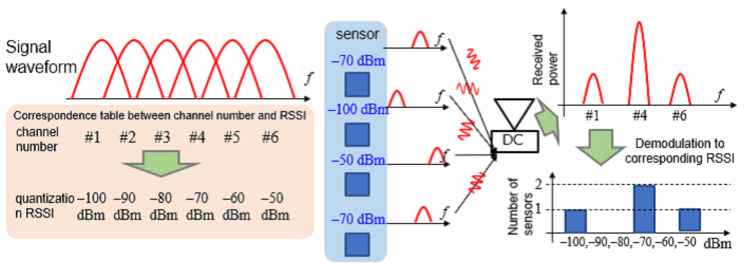
Physical Wireless Parameter Conversion Sensor Networks(PhyC-SNs).

**Figure 2 sensors-23-03308-f002:**
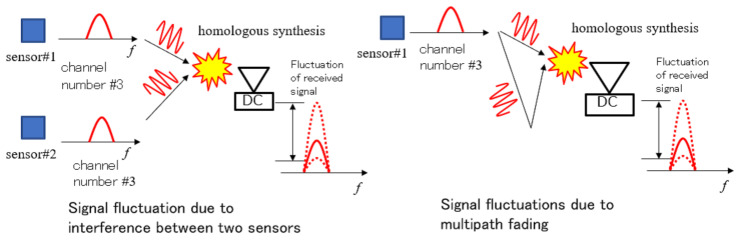
Detection Problem to Multiple Sensors in a PhyC-SN.

**Figure 3 sensors-23-03308-f003:**
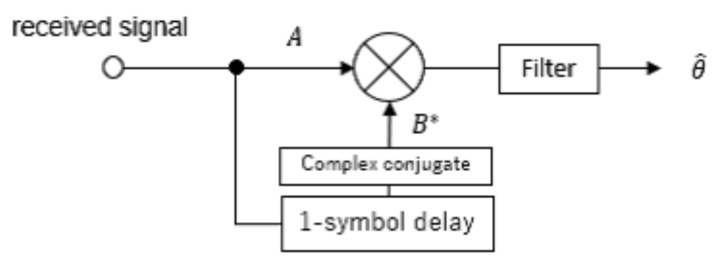
Delayed detection processing.

**Figure 4 sensors-23-03308-f004:**
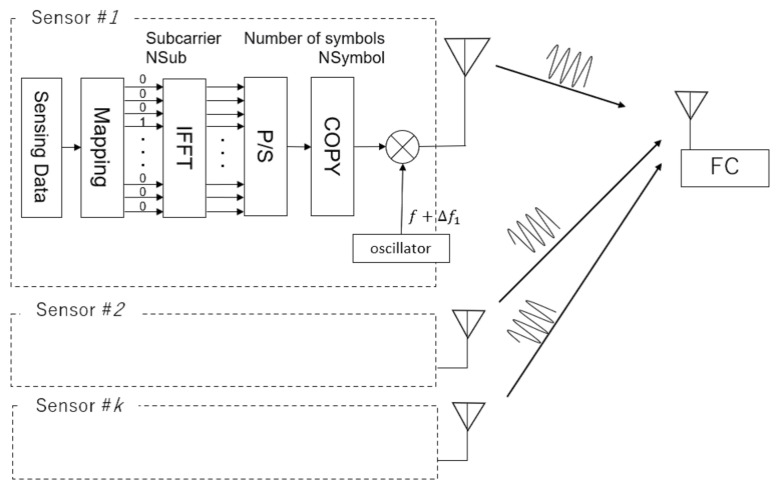
Transmitter side processing.

**Figure 5 sensors-23-03308-f005:**
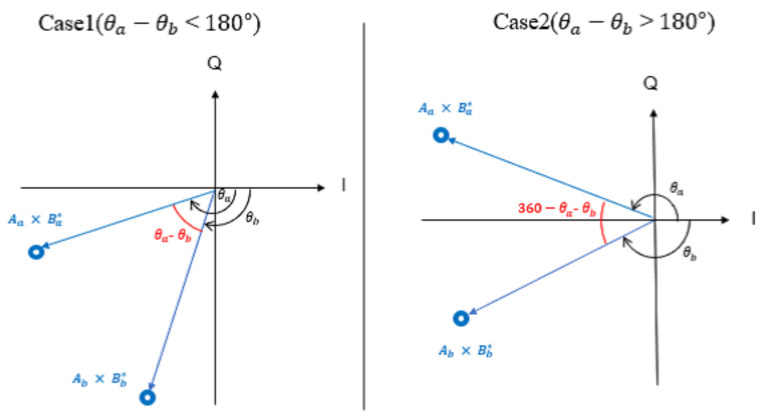
Calculation of the phase difference.

**Figure 6 sensors-23-03308-f006:**
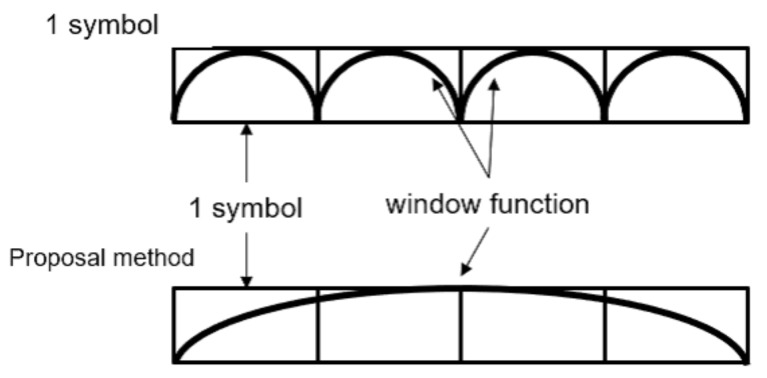
Application of the window function.

**Figure 7 sensors-23-03308-f007:**
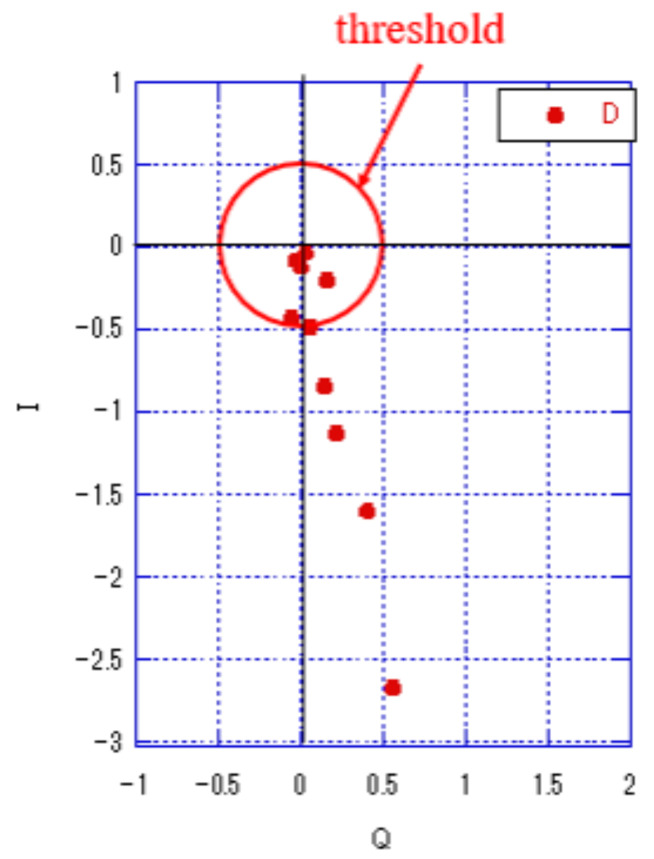
Receiver-side processing ("D" indicates the received signal point).

**Figure 8 sensors-23-03308-f008:**
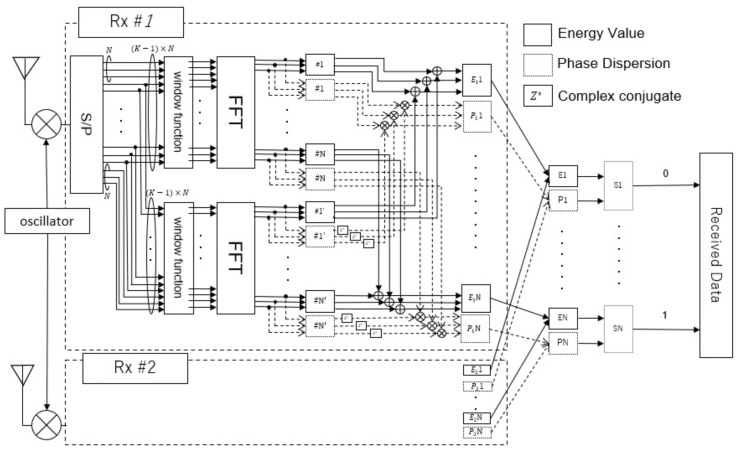
Receiver-side processing.

**Figure 9 sensors-23-03308-f009:**
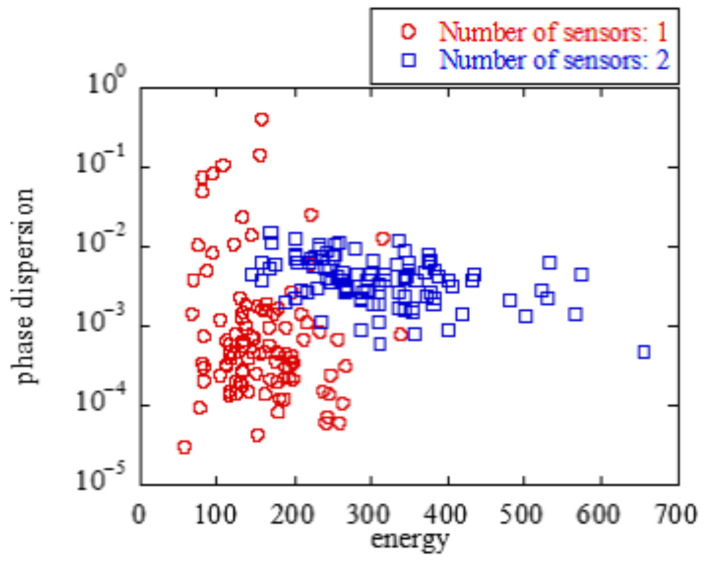
w/o antenna selection.

**Figure 10 sensors-23-03308-f010:**
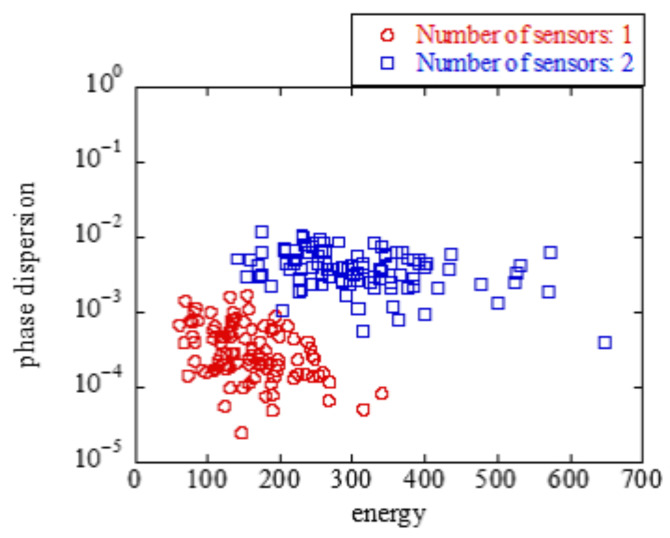
w/ antenna selection.

**Figure 11 sensors-23-03308-f011:**
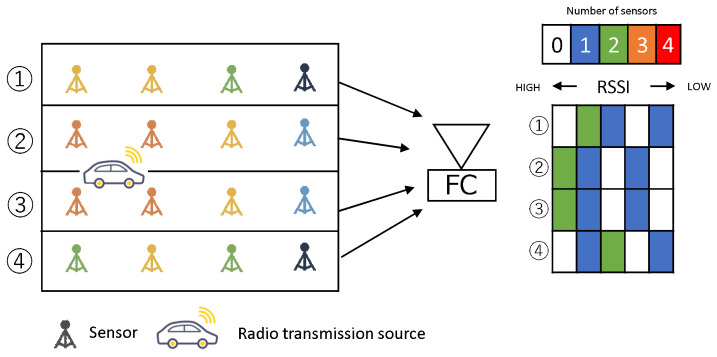
Aggregation Process.

**Figure 12 sensors-23-03308-f012:**
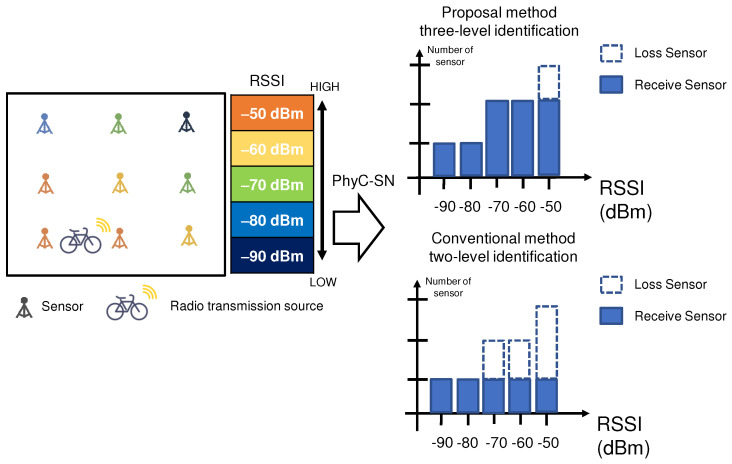
Comparison of missing sensor information.

**Figure 13 sensors-23-03308-f013:**
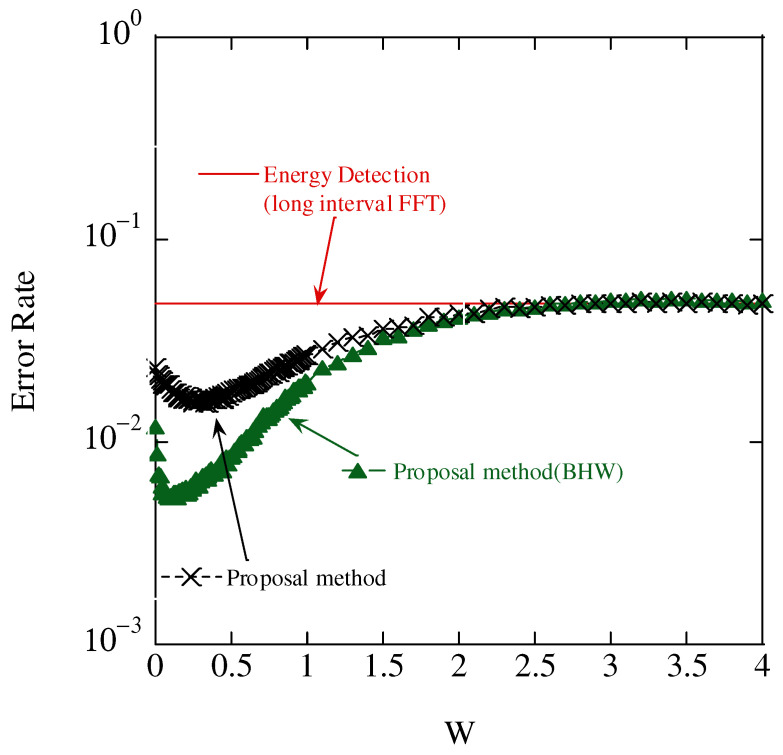
Aggregate accuracy results.

**Figure 14 sensors-23-03308-f014:**
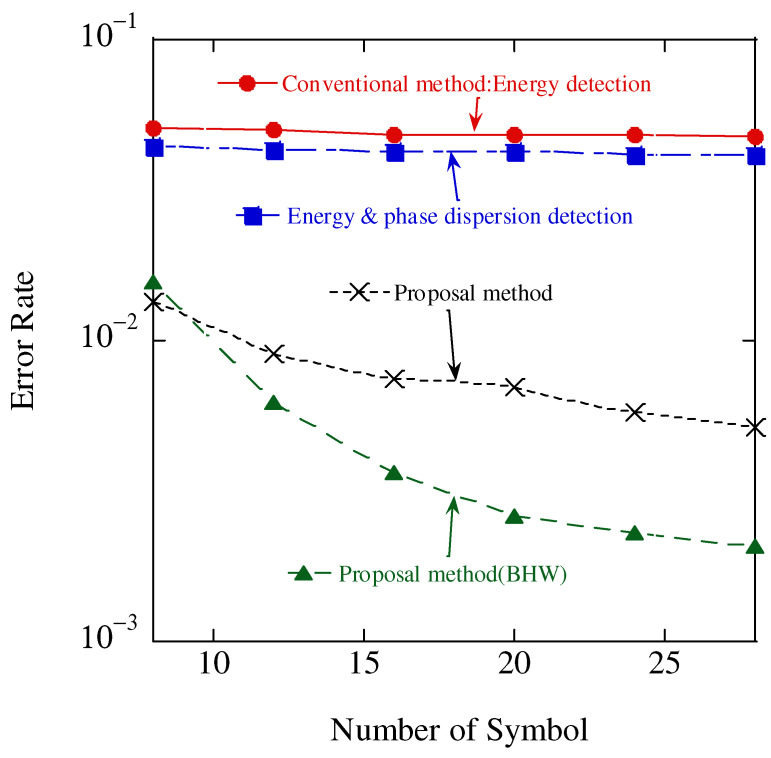
Aggregate accuracy results.

**Figure 15 sensors-23-03308-f015:**
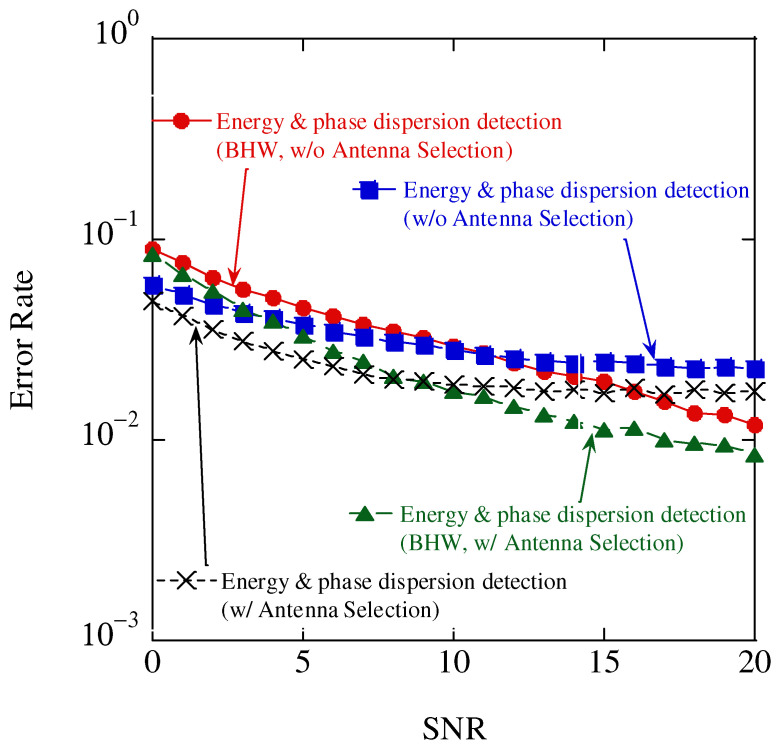
Aggregate accuracy results.

**Figure 16 sensors-23-03308-f016:**
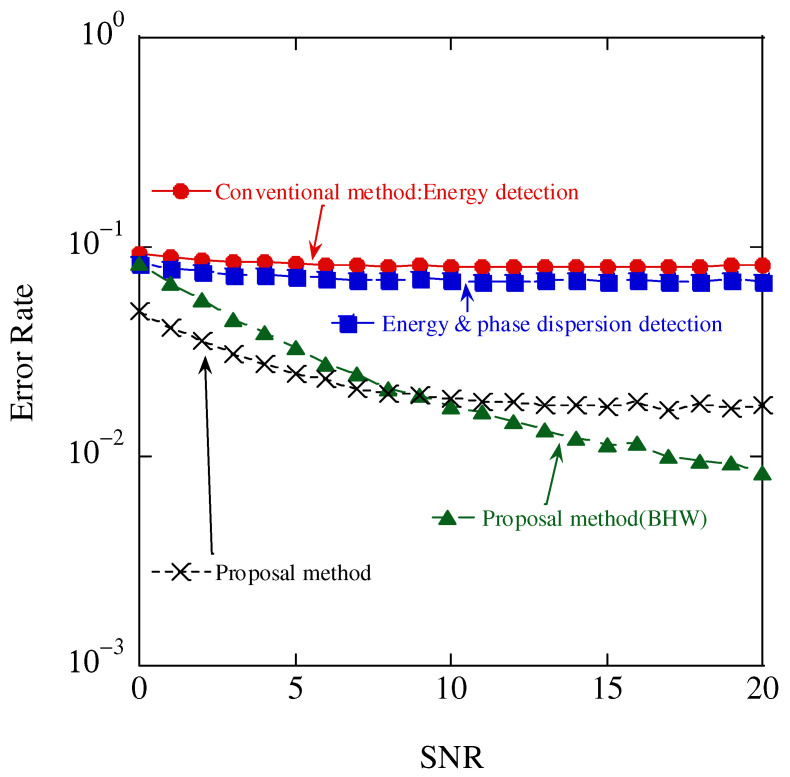
Aggregate accuracy results.

**Figure 17 sensors-23-03308-f017:**
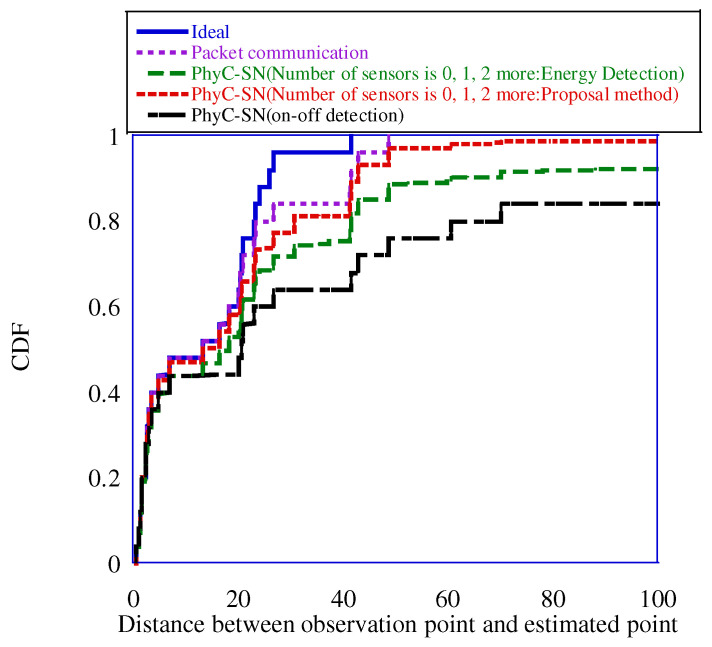
Positioning error at 16 channels.

**Figure 18 sensors-23-03308-f018:**
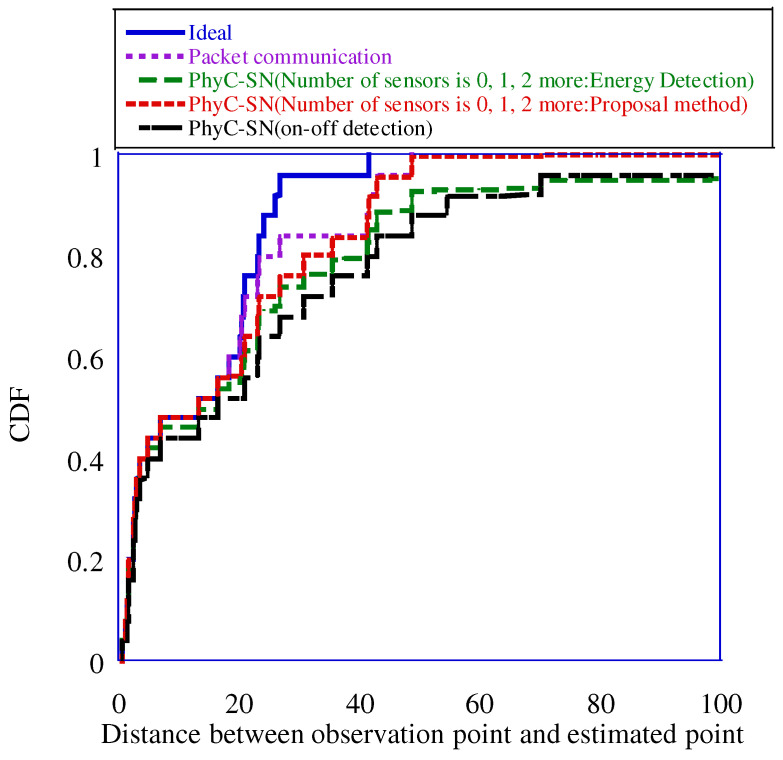
Positioning error at 64 channels.

**Table 1 sensors-23-03308-t001:** Simulation parameters.

Type of Data	Data
Number of receiving antennas	4
Fading environment	Rayleigh fading
Number of subcarriers	16
Frequency offset	[−0.4 0.4] independent
	uniform random numbers
Window functions	rectangular windows,
	Blackman-Harris windows
Number of sensors	35
Number of pre-training data for each sensor	60
Number of data for validation for each sensor	64

**Table 2 sensors-23-03308-t002:** simulation parameters part2.

Type of Data	Data
Size	800 m × 800 m
Number of sensors	136
Number of area divisions	8
Preliminary study points	37
Location Estimation Point	25
Central frequency	920.6 MHz
Bandwidth	0.125 MHz
Transmission power	10 dBm

**Table 3 sensors-23-03308-t003:** Comparison of identification errors.

PhyC-SN (Number of Sensors is 0, 1, 2 More)	16 Channels	64 Channels
Error Rate (Energy Detection)	0.1254	0.0380
Error Rate (Proposal method)	0.0136	0.0026

## Data Availability

Not applicable.

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
