# Peer review of "Three Level Recognition Based on the Average of the Phase Differences in Physical Wireless Parameter Conversion Sensor Networks and Its Effect to Localization with RSSI"

_sensors, 2023, doi:10.3390/s23063308_

Round 1

Reviewer 1 Report

In this paper the authors propose a novel approach to offer identification of concurrently transmitted sensors as well as radio source localization services, relying solely on basic physical layer information/modalities and mainly RSSI measurements.

The objective is certainty of high importance and impact as well as challenging since the whole approach is relying to phy layer parameters and characteristics manipulation and processing.

The use of English is acceptable although another round of review is suggested for minor wording mainly errors or better selections.

On the positive aspects the approach is described adequately and from different perspectives (i.e. the basic parameter conversion, how collision is detected, how the window function comes into play, how the detection sensitivity is enhanced, the use of SVM etc.) which attributed to the comprehensiveness of the work. Which increases the potentially interested reading audience.

On the weak points, a general comment is that the results section both simulated and real measurements should/could be more analytical and the graphs better and in more depth explained. This needs to be addressed in my opinion and will increase the added value of the paper.

On specific issues:

The title is a quite confusing. If possible to change it would be nice to be more descriptive of what the methods tries to do (localization and concurrent transmitting nodes identification) and how, based solely in phy layer parameters.

Lines 205-208…. I don’t the follow what was supposed to be there. Please elaborate or remove.

Line 257 “where is the” something is missing.

Line 297-298 please elaborate on why this assumption is made and how this affects or not the objectivity of the study.

Section 4.3 “Antenna Selection” not clear what it does and how it does it. Please elaborate.

Figure 12. not clear. Please elaborate on what is shown here.

Section 6. Please elaborate on the network setup in terms of topology, density and application rate and relative position between transmitters and receivers.

Line 425. what do you mean simulated. Also please provide more info on the devices, sensors or/and embedded system used. Please elaborate.

Section 6.2 Which simulation environment or network simulator was used ?

Line 498, what do you mean "radio sources" ?

And finally a general question/query. It would be very interesting if the authors could discuss somewhere in the document whether the processing can be done real-time on a real life embedded platform found e.g on a typical IoT device?

Overall the work is very interesting so if the paper can address the above comments the paper can certainly be revised and reconsidered.

Reviewer 2 Report

The paper deals with Three-level detection using the collision detection method in physical wireless parameter conversion sensor networks. The paper is logically divided into several basic chapters and subsections. 

The topic is interesting and I think it could be of interest to many readers.

Reviewer 3 Report

This paper proposes a collision detection method for physical wireless parameter conversion sensor networks (PhyC-SN) that can precisely identify the cases where more than one sensor reports the same sensor information. And the authors applies the proposed method to the position fingerprinting method, which measures the position of a radio source using the observation results of many sensors. In addition, The evaluations performed through computer simulations show that the proposed PhyC-SN achieves better positioning accuracy than the conventional energy detection method owing to improved accuracy of identifying the number of sensors. However, I have the following comments.

(1) In the introduction, the author can summarize the main contributions and elaborate the specific content of the rest of the article.

(2) When introducing the relevant studies, the authors can simply explain how they learned and improved their own PhyC-SN method from the previous research schemes.

(3) The image lines drawn by the authors through computer simulation are relatively fuzzy, and are expected to be improved.

(4) It is suggested to add the following references in the introduction, SWIPT Cooperative Spectrum Sharing for 6G-Enabled Cognitive IoT Network, Collaborative energy and information transfer in green wireless sensor networks for smart cities.

(5) The authors should proof read the paper to further improve the writing.

Reviewer 4 Report

This is an interesting Job in that the phase variance value focusing on the sensor-specific frequency offset can be used as an identifier to efficiently identifier or detect the sensor sending the data. However, 2 questions is needed to be answered before acceptance for publication:

1. To clarify the difference in the main idea and technical processing between the previous job also by the authors( Ref [13]) and the current submission;

2. To polish English expressions.

In addition, some leftover from the template are to be located and cancelled.

Round 2

Reviewer 1 Report

All the reviewer’s comments have been adequately and fully addressed. The additions required are satisfied increasing the clarity and thus interest in the work. The new title is also adding to enhancing the interested targe audience of the work so it is also welcomed.

Thus all my concerns are lifted.